# Differences in Emotional Conflict Processing between High and Low Mindfulness Adolescents: An ERP Study

**DOI:** 10.3390/ijerph19052891

**Published:** 2022-03-02

**Authors:** Xiaomin Chen, Xinmei Deng

**Affiliations:** 1School of Psychology, Normal College, Shenzhen University, Shenzhen 518060, China; 2060482047@email.szu.edu.cn; 2Center for Mental Health, Shenzhen University, Shenzhen 518060, China

**Keywords:** mindfulness, adolescent, face Stroop task, inhibitory control

## Abstract

Mindfulness is a state of concentration that allows individuals to focus on their feelings and thoughts without judgment. However, little is known regarding the underlying neural processes of mindfulness. This study used ERPs to investigate the differences between high and low trait mindfulness adolescents during emotional conflict processing. Nineteen low mindfulness adolescents (LMSs) and sixteen high mindfulness adolescent (HMSs) individuals were asked to complete a face Stroop task. The task superimposed emotional words on emotional faces to generate congruent (CC) and incongruent (IC) conditions. Continuous electroencephalogram data were recorded during the face Stroop task. Results revealed that for N450, the interaction of congruency and group was significant. The incongruent trials evoked a larger N450 than the congruent trials in the HMSs, whereas there were no significant differences between the two conditions in the LMSs. There were significant main effects of congruency for SP (slow potential). The incongruent trials evoked a larger SP than the congruent trials. The results suggest that mindfulness may only affect early conflict monitoring rather than later conflict resolution. The findings expand the neural basis of the effect of mindfulness on inhibitory control.

## 1. Introduction

Mindfulness has an impact on individual emotions, such as by reducing pressure [1], lowering negative emotions [2], and improving positive emotions [3], etc. ERP is widely used in affective neuroscience. Previous research has shown that adolescence is fraught with challenges, such as significant psychology fluctuation, greater social stress and unstable interpersonal relationships [4]. However, constant brain development makes it difficult for teenagers to regulate their emotions [5], so they are more likely to suffer from emotional dysregulation [6]. Examining the underlying mechanisms of mindfulness and emotional response through ERP may help adolescents survive this stormy period.

### 1.1. Mindfulness

Mindfulness is a state of concentration that allows individuals to focus on their feelings and thoughts without judgment [7]. Mindfulness is generally thought of as having two components: (1) focus attention on current thoughts and feelings, rather than shifting attention to other time [8] and (2) the psychological attitude of accepting the current experience [9], in which they do not immerse their attention in negative thoughts [8]. Mindfulness level can be divided into the trait mindfulness and state mindfulness [10]. Trait mindfulness refer to the state of mindfulness that individuals exhibit in daily lives; state mindfulness reflects an individual’s state of being mindful at one point in time [11].

Mindfulness can improve individual’s mental health, cognitive function and interpersonal interaction [12,13,14]. Especially in terms of emotional response and adjustment, self-report and neuroscience techniques confirmed that mindfulness played an important role [15]. Previous studies have shown that adolescents who are more mindful experience less anxiety [16]. Individuals with higher levels of mindfulness are associated with lower negative emotions and higher positive emotions [2,3].

### 1.2. Mindfulness and Inhibitory Control

Although the existing research has different explanations for the mechanism of the impact of mindfulness on emotion, previous research has shown that mindfulness reduces personal pain by not judging current thoughts and feelings [17,18,19]. In addition, studies using neuroimaging methods have found that functional neural flexibility modulates the relationship between mindfulness and anxiety symptoms, and greater neural flexibility may protect individuals from being trapped in negative emotions [20]. However, several studies have shown that mindfulness practice can improve individual inhibitory control ability, thus improving individual emotional response and emotional regulation ability [21].

Inhibitory control is to identify the existence of conflict in cognitive processing and resolve conflict to achieve the desired goal [22]. Inhibitory control is a very important part of cognitive control, through which individuals can regulate cognitive processing [21]. In addition, there are two types of inhibitory control, active inhibition and reactive inhibition respectively [23]. Among them, reactive inhibition is a response to an external stimulus and is considered to be a suspension of an ongoing response [24]. Neuroscience studies have shown that reactive inhibition is related to frontal lobe activation. However, the exact location in the frontal cortex seems to be affected by the difficulty of the task [25]. Specifically, the right prefrontal–parietal circuits were affected by task characteristics, while the pre supplementary motor area was only linked to reactive inhibition [23].

Several studies have found that mindful states can successfully reflect inhibitory control [26]. Heeren and colleagues found that subjects who received mindfulness practices performed better on inhibition control tasks [27]. In addition, a group of individuals with attention disorders who received mindfulness practices made fewer errors on cognitive control tasks [28]. The effect of mindfulness on the process of emotional inhibition occurs through changes in the brain structures and circuits that are closely related to it [21]. Changes in brain structures include greater activation in the medial prefrontal cortex [29]. Moreover, the amygdala is also more tightly connected to the ventromedial prefrontal cortex [29].

The classic Stroop task [30] could measure inhibitory control ability in conflict conditions [31], but this is not able to measure the cognitive inhibition caused by emotional distractors [32]. Therefore, this study adopted the face Stroop task. In a previous study using the face Stroop task, participants were asked to recognize the emotional expression of a facial expression while ignoring words that were consistent or inconsistent with the facial expression [29]. Response conflict is caused by the emotional inconsistency of the experimental stimulus [31]. Past research has shown that an important mechanism helps to control the conflicts caused by the face Stroop task [33].

### 1.3. ERP and Inhibitory Control

ERP has high time resolution and can record the cognitive process of executive control [34]. A large number of ERP studies have found that two ERP components, N450 and SP, can distinguish trial consistency [35,36]. N450 is a negativity of the central frontal region that peaks around 400 ms after stimulus onset [37]. Source location suggests that N450 is located in ACC [38]. Existing studies have generally led to conclusions that N450 is relate to the conflict monitoring process [39].

SP began about 500 ms after the onset of stimulation [40]. Source localization suggests that it may be associated with activation of the frontal, parietal and occipital lobes. Several studies have shown that SP is more positive in the inconsistent trials than in the consistent trials [41].

### 1.4. The Present Study

During this important period, it is important for adolescents to master effective methods of emotional regulation. Emotional regulation is an individual’s response to emotions in order to make appropriate responses to environmental requirements. Mindfulness and emotional inhibition are common emotional regulation strategies in daily life [42]. Emotional inhibition can change the expression behavior of individuals, but it could not make individuals free from negative emotions, and will consume a lot of cognitive resources. Mindfulness is the antithesis of such admittedly problematic strategies for suppressing emotions. It emphasizes the full acceptance of all emotional experiences by individuals, and it changes the relationship between individuals and events [43]. However, little is known about the neural mechanisms underlying the impact of mindfulness on emotion. In this study, the ERP method was used to explore the differences between high and low trait mindfulness individuals in emotional conflict experiments. In addition, mindfulness has a positive effect on individual emotional inhibitory control [21]. Mindfulness allows people to focus and reduce distractions from extraneous things. When attention is distracted, the individual becomes aware and refocuses his attention on the task he is performing. In other words, people with high levels of mindfulness are good at refocusing their attention from a conflict situation [44]. Therefore, we hypothesized that high trait mindfulness adolescents would show smaller conflict effects than low trait mindfulness adolescents.

## 2. Methods

### 2.1. Sample

According to the prior power analysis of G* Power software [45], 22 participants were required to detect the effect size (0.4) in a 2 × 2 repeated ANOVAs with 80% power. Thus, we recruited 38 adolescent participants in the present study. All the teenagers are from Shenzhen, China. Participants and their parents signed informed consent. All participants had normal or corrected-to-normal vision, were right-handed. No participants reported a history of psychiatric or neurological disorders.

Participants were selected based on their scores on the Chinese version of the 39-item FFMQ. The Chinese version of the FFMQ assesses the level of mindful skills in five facets [46]. These five factors are observation (e.g., when I take a shower, I pay attention to the sensation of water running over my body; α = 0.75), description (e.g., I’m good at putting my feelings into words; α = 0.84), acting with awareness (e.g., when I’m doing something, my mind often wanders and I’m easily distracted; α = 0.79), non-judging (e.g., I blame myself for having irrational or inappropriate emotions; α = 0.66), non-reacting (e.g., I feel my emotions, but I don’t have to react to them; α = 0.45). Items are rated on a 5-point Likert scale ("not at all" to" completely true"). Cronbach’s alpha of each subscale of CH-FFMQ is acceptable [46]. We averaged across the 39 items (after reverse scoring the relevant items) to create the total s trait mindfulness (α = 0.73).

According to the grouping methods of previous studies [47,48], we used a median split at trait mindfulness score to define low and high mindfulness adolescent groups. Three participants were excluded from analyses due to excessive EEG artifacts and technical error. Accordingly, there were 19 low mindfulness adolescents (LMS; 12 male and 7 female, aged from 10 to 14 years old, *M*age = 12.44, *SD* = 1.35) and 16 high mindfulness adolescents (HMS; 10 male and 6 female, aged from 10 to 14 years, *M*age = 12.43, *SD* = 1.35). HMSs (*M*_HMSs_ = 129.73, *SD* = 10.54) and LMSs (*M*_LMSs_ = 109.02, *SD* = 7.15) had significant differences in overall trait mindfulness scores; *t* = −6.90, *p*= 0.000*,* Cohen’s *d* = 2.30. The sample size of this study was also in line with a typical ERP study [49,50,51]. 45.71% of the teenagers in the sample were the only child. 82.85% of fathers and 82.85% of mothers had college degrees. The experiment was carried out in accordance with the guidelines of the Declaration of Helsinki and approved by the local Institutional Review Board.

### 2.2. Stimuli

Different emotional faces and emotional words were selected. Forty emotional faces were selected from the Chinese Facial Affective Picture System [52]. The experiment consisted of two types of pictures. One type of picture is negative faces (10 male and 10 female; identity: *M* ± SD = 81.09 ± 5.97; arousal: *M* ± SD = 5.86 ± 0.77). Another type of picture is neutral faces (10 male and 10 female; identity: *M* ± SD = 84.23 ± 6.70; arousal: *M* ± SD = 5.77 ± 0.21). In all, 20 negative and 20 neutral faces were selected. There was no significant difference between negative and neutral faces in identity ratings or arousal ratings (*p*> 0.05). All of the words were selected from the Chinese Affective Words System [53]. The experiment consisted of two types of words. One type of word is negative words (10 nouns; valence _*M* ± SD_ = 2.84 ± 0.15; arousal _*M* ± SD_ = 6.23 ± 0.24). Another type of word is neutral words (10 nouns; valence _*M* ± SD_ = 5.27 ± 0.17; arousal _*M* ± SD_ = 4.16 ± 0.55). In all, 20 negative and 20 neutral words were selected. Independent sample t-tests were used to compare the valence and arousal between negative and neutral words. Results showed that there were significant differences between negative and neutral words in identity ratings or arousal ratings. The negative words were more arousing and less pleasing than the neutral words (both *p* < 0.001). 

The red words overlay every facial expression. Combine two different types of faces and words into consistent and inconsistent conditions. Specifically, the conflict condition was that the facial expression was inconsistent with the emotional valence of the red words; the reverse was the non-conflict condition. The font size of the word is 48 Song. Combined stimuli displayed on a 19-inch monitor. The subjects viewed the stimulus at a distance of about 60 cm.

### 2.3. Procedures

We employed a 2 (group: HMSs vs. LMSs) × 2 (congruency: congruent vs. incongruent) two-factor mixed design. After the participants filled out basic demographic information, the face Stroop task began (see Figure 1) [54]. Participants were asked to respond accurately and quickly to facial expressions. At the beginning of each trial, participants were presented with fixation "+" for 500 ms, followed by a random blank screen lasting 300-500 ms. The stimulus then appeared in the center of the screen for 1500 ms, followed by a random blank screen lasting 800 or 1200 ms. When the participant’s reaction time (RT) exceeded the time limit set by the program (1500 ms), it was considered a false response.

All 80 images were presented twice in the face judgment task. The experimental conditions are composed of congruent conditions (CC) and incongruent conditions (IC). There were 2 blocks of 80 trials, resulting in 160 trials in total. The order of the trials within each block was random. The experiment lasted 10-15 minutes for each participant.

### 2.4. EEG Recording, Data Collection, and Analysis

The EEG data were recorded from a 64-channel amplifier (BrainAmp, Brain Products, Germany) based on the international 10/20 system. During the recording process, the sampling rate was 500Hz and the electrode impedance was controlled below 5 kΩ. The EEG data were further processed by Brain Vision Analyzer software and were filtered by a bandpass of 0.5–30 Hz [55]. The independent component analysis (ICA) algorithm was used to remove artifacts. The continuous data were divided into 1200 ms epochs, with the 200 ms epochs before stimulation as baseline. The wrong responses of the subjects were not included in the subsequent analysis. At least 41 trials were available in the 2 conditions for each participant. The EEG signals with artifacts greater than ±80μV was not included in subsequent analysis. The final mean number of valid epochs was 65.60 (82.00%) for CC, 60.11 (75.14%) for IC.

Based on the waveform of this study and previous studies [36,56], two obvious ERP components, N450 and SP, were analyzed within the time windows of 380–460 and 650–850, respectively. For N450, the average amplitude of the frontal electrode (Fz) was calculated [57,58]. In addition, the average amplitude of the parietal sites (Pz) was analyzed for SP [58].

Two-way repeated measures ANOVAs with 2 groups (HMS and LMS) × 2 congruency levels (congruent and incongruent) were analyzed for N450 and SP amplitudes, respectively. SPSS 20.0 was used for statistical analysis. The significance level was set as *p* < 0.05. If the assumption of sphericity was not met, greenhouse-Gaither correction was used. The Bonferroni test was used for *post hoc* tests.

## 3. Results

***N450*.** For N450, The main effect of congruency was not significant: *F*(1,33) = 2.96, *p* = 0.095, η_p_^2^ = 0.08. The main effect of group was also not significant: *F*(1,33) =1.84, *p* = 0.184, η_p_^2^ = 0.05. 

The interaction of congruency and group was significant: *F*(1,33) = 4.43, *p* = 0.043, η_p_^2^ = 0.12 (Figure 2 and Figure 3). Post hoc tests found that the incongruent trials evoked a larger N450 than the congruent trials in the HMSs (*p* = 0.014), whereas there were no significant differences between the two types of condition in the LMSs (*p* = 0.779) (Table 1).

*Slow potential (SP).* For SP, the main effect of congruency was significant: *F*(1,33) = 11.13, *p* = 0.002, η_p_^2^ = 0.25. Incongruent trials produced larger SP amplitude than congruent trials (*p* = 0.002). The main effect of group was not significant: *F*(1,33) =3.10, *p* = 0.088, η_p_^2^ = 0.09. The interaction of congruency and group was also not significant: *F*(1,33) = 0.36, *p* = 0.552, η_p_^2^ = 0.01 (see Figure 2 and Figure 4).

## 4. Discussion

Mindfulness has positive effects on individual cognitive function [59]. For example, some researchers have found that mindfulness practices in schools can improve children’s inhibitory control [60]. However, to our knowledge, there is little neuroscientific evidence to describe the mechanism by which mindfulness affects individual inhibitory control. The present study explored the differences in inhibition control ability of individuals with different trait mindfulness levels in a Stroop task. By identifying the relationship between trait mindfulness and inhibitory control, we have the opportunity to elucidate the neural basis of mindfulness’s influence on inhibitory control.

N450 is a negativity of the central frontal region that peaks around 400 ms after onset [61]. In this study, the main congruent effect of N450 was not significant, which may be due to the emotional conflict task adopted in this study. N450 reflects the automatic process of conflict detection, so it is vulnerable to the adverse impact of emotional stimulus changes and consumes more cognitive resources, resulting in a weakened conflict monitoring of the stimulus [62]. Therefore, no congruent effect of N450 was found in this study. In addition, adolescents have not developed the mature brain functions [63]. The low mindfulness group might need to invest more cognitive resources under different conditions. However, the high mindfulness group could be more able to complete the task by investing less cognitive resources in different conditions because they were better able to keep their attention focused on the target event while minimizing the degree of distraction [17]. Therefore, there was no significant difference between groups.

The current findings also indicate that the incongruent trials evoked a larger N450 than the congruent trials in the HMSs, whereas there were no significant differences between the two conditions in the LMSs. N450 reflects conflict detection and is related to the amount of resources that individuals expend on execution control [36]. The N450 induced by HMSs in the incongruent trials was larger than that in the congruent trials, suggesting that mindfulness is particularly important for the improvement of individuals’ conflict suppression ability. Malinowski et al. found that mindfulness practice improved individual performance on cognitive control tasks [64]. Kelly et al. found that attentional exercise also made individuals perform better on cognitive control tasks [65]. However, adolescents have not developed the mature brain functions [63]. The low mindfulness group had to devote just as many cognitive resources to the congruent stimulus as they did to the incongruent stimulus, which could the reason that the low mindfulness group showed no congruent effect on the emotional conflict task.

In addition, the incongruent trials produced larger SP amplitudes than the congruent trials. In line with previous studies, the conflict slow potential is positivity in the middle parietal lobe and is associated with increased neural resources [58]. It may reflect the process in which subjects recognized facial expressions and inhibited interfering words. However, no interaction between consistency and group was found in this study, which may be caused by differences between different conflict types [66]. Several studies have shown that emotional conflict control occurs earlier than cognitive conflict control [66]. In the early stage of conflict detection, adolescents may not accumulate enough neural resources due to the adjustment of emotional stimuli, which makes it difficult for subjects to solve emotional conflict [67]. Therefore, in contrast to the early effort of conflict monitoring cognitive resources, both the high mindfulness group and the low mindfulness group put the same amount of effort into congruent and incongruent conditions during late processing. This suggests that mindfulness may only affect early conflict monitoring rather than later conflict resolution.

It was found that the high mindfulness group induced a larger N450 in the inconsistent condition, whereas the low mindfulness group did not show a consistent effect. N450 reflects the monitoring of conflict processes [68]. The greater N450 reflects that more cognitive resources are recruited in conflict detection. Therefore, the N450 results suggest that mindfulness improves the ability to monitor emotions in early adolescence. In addition, no interaction between group and consistency was found for SP in this study. SP usually reflects conflict resolution in the later stage [35]. The combined results of N450 and SP suggest that the effect of mindfulness on adolescent inhibitory control is mainly reflected in early conflict monitoring rather than later conflict resolution.

## 5. Limitations and Future Directions

This study explored the effect of mindfulness on adolescent inhibitory control. However, previous neuroscience studies have found that adolescents may show different neural activation and behavioral responses from adults during conflict. As the neural mechanism involved in attentional control is not fully developed, adolescents’ conflict adaptation ability is considered to be poor [69]. In addition, other studies have found that adolescents activate more cognitive resources for lower levels of conflict, whereas adults do the opposite [70]. In view of the above findings, it is possible to study the influence of mindfulness on adult inhibitory control in the future and further explore the similarities and differences between the influence of mindfulness on adolescent and adult inhibitory control.

In addition, due to the priority of negative stimuli in cognitive processing for survival and development, current studies mainly focus on negative and neutral stimuli. However, previous studies have found that positive emotions can boost performance, especially in flexible task situations [71]. It’s worth noting that positive emotions may impair cognitive control because individuals may rely more on heuristics [62]. Future studies can add positive emotional stimuli to further explore the similarities and differences of the effects of mindfulness on individuals’ processing of different emotional materials.

## 6. Conclusions

This study examined the effects of mindfulness on adolescent inhibitory control and expands the neural basis of mindfulness for adolescent inhibitory control. We found that the high mindfulness group induced a larger N450 in the inconsistent condition, whereas the low mindfulness group did not show a consistent effect. These findings suggest that mindfulness may be able to improve adolescents’ ability to monitor emotions in the early stage. In addition, this study found that SP did not show a significant interaction between group and consistency. This may indicate that the influence of mindfulness on adolescent inhibitory control is mainly reflected in the early stage of conflict monitoring rather than in the later stage of conflict resolution. These findings help to clarify the mechanisms by which mindfulness promotes inhibitory control in adolescents.

## Figures and Tables

**Figure 1 ijerph-19-02891-f001:**
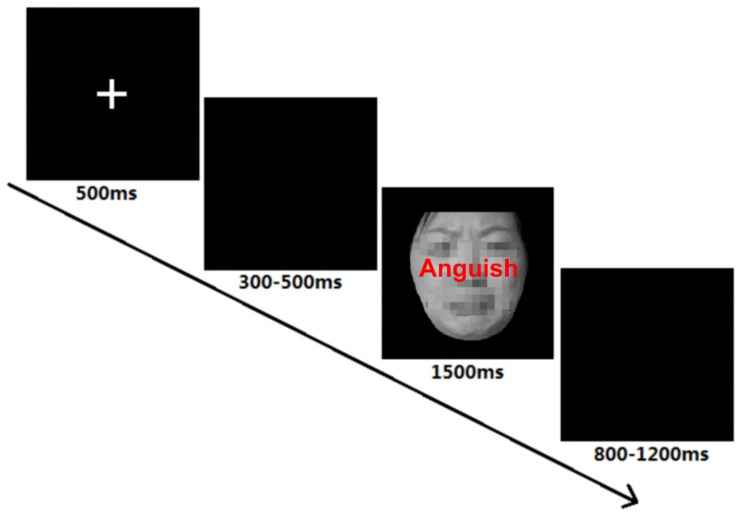
Procedure of each trial.

**Figure 2 ijerph-19-02891-f002:**
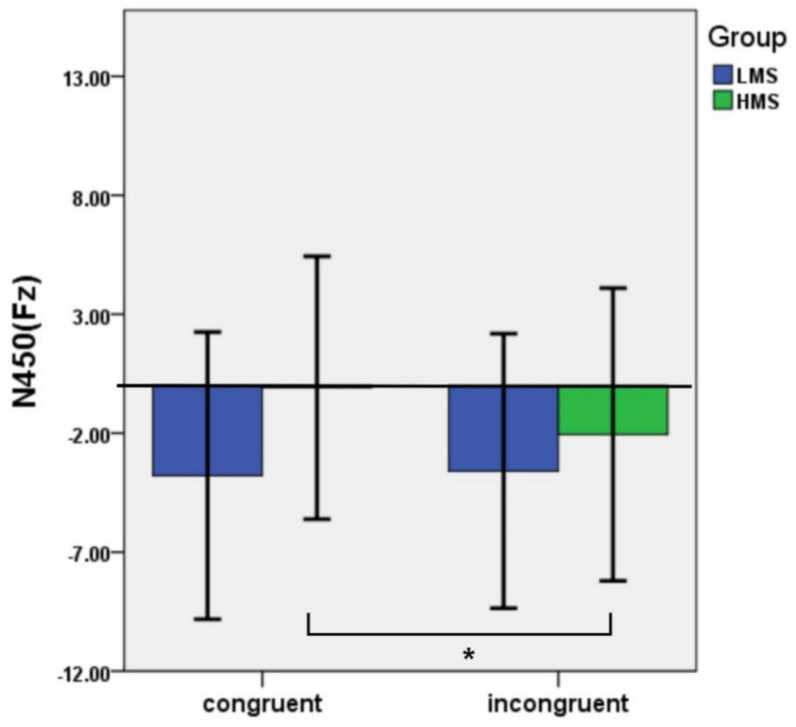
Average amplitudes of N450 and SP in different conditions between LMSs and HMSs. Note. **p*<.05; ***p*<.01.

**Figure 3 ijerph-19-02891-f003:**
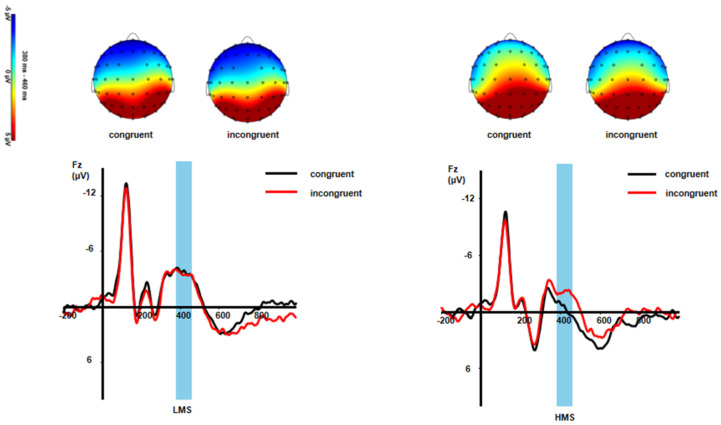
Grand-averaged N450 waveforms and topographic maps of scalp voltage in mean activity in response to different conditions between LMSs and HMSs.

**Figure 4 ijerph-19-02891-f004:**
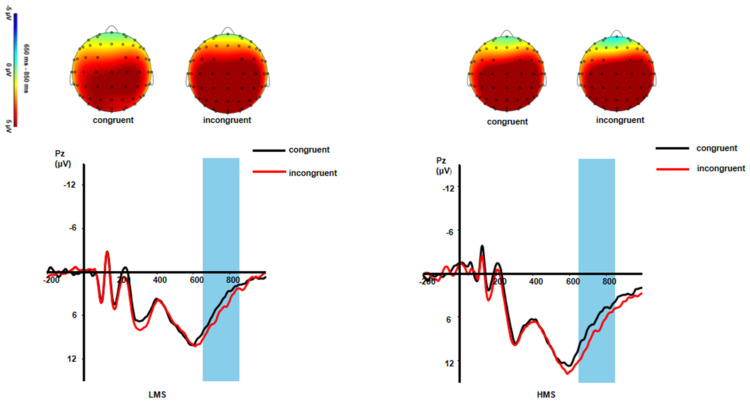
Grand-averaged SP waveforms and topographic maps of scalp voltage in mean activity in response to different conditions between LMSs and HMSs.

**Table 1 ijerph-19-02891-t001:** Average amplitudes of N450 and SP in different conditions between LMSs and HMSs.

Component	Group	Congruent	Incongruent	*t*	*d*	*p*	95%CI	
		(*M* ± *SD*)	(*M* ± *SD*)					
N450	LMSs	−3.79 ± 6.03	−3.59 ± 5.77	−0.31	−0.03	0.757	−1.51	1.12
	HMSs	−0.09 ± 5.52	−2.06 ± 6.15	2.35	0.34	0.033	0.18	3.75
SP	LMSs	4.50 ± 4.06	5.58 ± 4.30	−1.87	−0.26	0.077	−2.29	0.13
	HMSs	6.72 ± 4.34	8.27 ± 4.40	−2.99	−0.35	0.009	−2.66	−0.45

LMSs = low mindfulness adolescents; HMSs = high mindfulness adolescents.

## Data Availability

The data used in this study are available upon reasonable request from the corresponding author.

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
