# Peer review of "Differences in Emotional Conflict Processing between High and Low Mindfulness Adolescents: An ERP Study"

_ijerph, 2022, doi:10.3390/ijerph19052891_

Round 1

Reviewer 1 Report

This is a very interesting study using neuroscience to investiagte the use of mindfulness in adolesence inhibitory control.

The one are that could be improved is the background, where more could be written when explaining previous studies, theories around the topic.

Reviewer 2 Report

Dear Editor,

in the manuscript entitled “Differences in the Emotional Conflict Processing between High and Low Mindfulness Adolescents: An ERP Study”, the authors enrolled 19 low mindfulness adolescents (LMSs) and 16 high mindfulness adolescents (HMSs) individuals and asked them to perform a Face Stroop task with congruent (CC) and incongruent conditions (IC). Results showed that incongruent trials evoked a larger N450 than the congruent trials in the HMSs participants. Additionally incongruent trials evoked larger slow potentials than congruent trials. Authors speculate that mindfulness affects the inhibitory control ability of emotional conflict, but not the inhibition ability of cognitive conflict.

    This is a well-conceived and interesting study. In my opinion the paper is of potential interest for publication on “The International Journal of Environmental Research and Public Health”.

For what concerns the manuscript these are my comments:

1. Overall, the introduction is not very smooth. Many concepts are repeated several times in the text but they are not well explained.

I believe that authors should make an effort to make the text more readable, clear and enjoyable by a wide audience such as that of IJERPH.

2. The topic of the paper is mindfulness and inhibitory control, however inhibitory/cognitive control is not well introduced and the network observed in old and recent metanalyses, as well as the associated cognitive models are not reported and treated in the text, apart for this little piece of text prevalently related to the brain structure : “The effect of mindfulness on the process of controlling emotional inhibition occurs through changes in the brain structures and circuits that are closely related to it, for example, increasing gray matter in the inferior frontal gyrus [20].The changes in brain structures include activity in the medial prefrontal cortex [24]. Moreover, there are also stronger functional connections in the amygdala and ventromedial prefrontal cortex [25]”.

    Authors should describe better the neural network of cognitive control (see Simmonds et al., 2008 and Gavazzi et al., 2021 for a more recent work and model of cognitive control). I believe that the data interpretation may really improve by introducing relevant fMRI metanalyses on the neural network of  proactive and reactive cognitive control, especially by taking into consideration this point reported by authors in introduction: “SP began about 500ms after the onset of stimulation [37]. Source localization suggests that it may be caused by the activity of a distributed network involving the middle and lower frontal gyrus, parietal cortex, and occipital cortex

Relevant bibliography

Simmonds, D. J., Pekar, J. J., & Mostofsky, S. H. (2008). Meta-analysis of go/no-go tasks demonstrating that fMRI activation associated with response inhibition is task-dependent. Neuropsychologia.

Gavazzi, G.; Giovannelli, F.; Currò, T.; Mascalchi, M. Contiguity of proactive and reactive inhibitory brain areas: A cognitive model based on ALE meta-analyses. Brain Imaging Behav. 2020.

3. In Methods (Sample), authors wrote: “HMSs (MHMSs=129.73, SD=10.54) and LMSs (MLMSs=109.02, SD=7.15) had significant differences in overall trait mindfulness scores (p<.001).” Can the authors report the statistical test and not just the p-value?

4. In Methods (Stimuli), authors report: “In all, 20 negative and 20 neutral words were selected. Results of the t-tests showed that there were significant differences in valence and arousal between negative and neutral words. The neutral words were lessarousing and more pleasing than the negative words (both p < .001).”

    Which t-tests? the authors should add more details

5. Please improve the quality of Figure2.

6. It would be of interest (and would enrich also the paper) to add in discussion a brief speculation related to the potential role exerted by the neural correlates that can be responsible for the effects measured by authors.

Reviewer 3 Report

Thank you for the opportunity to review your work.

The purpose of this study was to investigate the differences between high and low trait mindfulness adolescents during emotional conflict processing using ERPs. The findings indicate that mindfulness affects the inhibitory control ability of emotional conflict, but not the inhibition ability of cognitive conflict, expanding the neural basis of the effect of mindfulness on inhibitory control. The paper was interesting to read, and I believe that with some major revisions, the paper can make a substantial contribution to the literature.

Introduction

- p. 3: In The Present Study section

You mentioned “emotion regulation” and particularly suppression. I think emotion regulation is a huge concept in emotion literature. You may want to elaborate on this a little more as I think your readers may find it helpful.

-Your hypothesis is introduced here: “Because of the positive effects of mindfulness on individual emotional suppression [20], we hypothesized that high-trait mindfulness adolescents would show smaller conflict effects than low-trait mindfulness adolescents.” However, rationales for this hypothesis are not explained enough. Please provide more details on how this hypothesis was established.

Method

-You used the FFMQ to measure trait mindfulness of the participants. Did you use the validated Chinese version? I suggest you provide more detail on this measure.

-You mentioned “The sample size of this study was also in line with a typical ERP study [42].” Although it is cited, it would be more informative if you introduce how the sample size is decided in a typical ERP study.

Discussion

In general, I feel the Discussion section is weak and pretty short. Could you provide more thorough discussions on the following results:

- “For N450, the main effect of congruency was not significant. The main effect of group was also not significant.” Please discuss some possible reasons for this result.

- “…there were no significant differences between the two types of conditions in the LMSs.” Discussion for this non-significant difference in the LMSs is missing. Could you discuss for this finding?

- Regarding the result on no interaction between consistency and group, you explained “…which could be attributed to differences between emotional conflict and cognitive conflict suppression control.” Could you elaborate on this?

*Also, please add limitations of the study and accordingly suggest future research directions.

Conclusion

I think you can extend this section. You mentioned “The findings expand the neural basis of the effect of mindfulness on inhibitory control.” Probably you can add more implications of your study, especially relating to the adolescents.  

Round 2

Reviewer 3 Report

Dear authors,

I found your edits to be very helpful in understanding your research further. Overall it appears that you have addressed the main issues highlighted in the first submission. The discussion and the final conclusion provide useful information for the readers.

Thank you for your hard work.